# The Socioeconomic Characteristics of Childhood Injuries in Regional Victoria, Australia: What the Missing Data Tells Us

**DOI:** 10.3390/ijerph18137005

**Published:** 2021-06-30

**Authors:** Blake Peck, Daniel Terry, Kate Kloot

**Affiliations:** 1School of Health, Federation University, Ballarat, VIC 3350, Australia; d.terry@federation.edu.au; 2School of Medicine, Deakin University, Warrnambool, VIC 3220, Australia; kate.kloot@deakin.edu.au

**Keywords:** childhood injury, emergency presentation, socioeconomic, administration database, rural

## Abstract

Background: Injury is the leading cause of death among those between 1–16 years of age in Australia. Studies have found that injury rates increase with socioeconomic disadvantage. Rural Urgent Care Centres (UCC) represent a key point of entry into the Victorian healthcare system for people living in smaller rural communities, often categorised as lower socio-economic groups. Emergency presentation data from UCCs is not routinely collated in government datasets. This study seeks to compare socioeconomic characteristics of children aged 0–14 attending a UCC to those who attend a 24-h Emergency Departments with an injury-related emergency presentation. This will inform gaps in our current understanding of the links between socioeconomic status and childhood injury in regional Victoria. Methods: A network of rural hospitals in South West Victoria, Australia provide ongoing detailed de-identified emergency presentation data as part of the Rural Acute Hospital Data Register (RAHDaR). Data from nine of these facilities was extracted and analysed for children (aged 0–14 years) with any principal injury-related diagnosis presenting between 1 February 2017 and 31 January 2020. Results: There were 10,137 injury-related emergency presentations of children aged between 0–14 years to a participating hospital. The relationship between socioeconomic status and injury was confirmed, with overall higher rates of child injury presentations from those residing in areas of Disadvantage. A large proportion (74.3%) of the children attending rural UCCs were also Disadvantaged. Contrary to previous research, the rate of injury amongst children from urban areas was significantly higher than their more rural counterparts. Conclusions: Findings support the notion that injury in Victoria differs according to socioeconomic status and suggest that targeted interventions for the reduction of injury should consider socioeconomic as well as geographical differences in the design of their programs.

## 1. Introduction

Childhood injury is the leading cause of death and ongoing disease burden globally [1]. In Australia the picture is no different, with injury being the leading cause of death among those between 1–16 years of age [2]. Injury is also the principal cause of hospitalisation in this age group [3], with long-term disability resulting in significant impacts for both children and their families [4]. National and international studies have found that rates of injury mortality increase with socioeconomic disadvantage [5,6,7]. In contrast, studies examining injury morbidity have been less consistent, with some reporting an inverse relationship [5,7,8] and others a positive trend [9]. In Australia the links between injury morbidity and socioeconomic status (SES) remain less clear.

Despite the ongoing investment in prevention strategies and the personal, familial, and broader social burdens that childhood injury pose, there has been little examination of the epidemiological characteristics of childhood injury in the context of SES in Victoria. Studies such as these are essential to understand the genuine injury burden, to inform policy makers of how best to prioritise injury prevention strategies, and to evaluate their effect over time [10].

The exception is a study by Mitchell, Curtis et al. [9] whose primary outcome measure was to determine the number, incidence and temporal trends of hospitalised injury involving children in Australia over a ten-year period. While these authors ultimately reported no change in the incidence of childhood injury hospitalisations in Australia between 2002 and 2012, they did identify fluctuations in the incidence of childhood injury with regard to SES. Children residing in areas of socioeconomic disadvantage showed a slightly higher proportion of injury across all age groups. Of interest, however, the authors acknowledged the absence of some 20% of the Victorian data from their analysis on the basis that the hospital records were unable to be linked.

In determining the existence of a relationship between socioeconomic disadvantage and childhood injuries, it follows that we need to examine the proportion of injury amongst children from areas of the state generally identified as the most disadvantaged: regional and rural Victoria [11]. It is well-documented that indices of SES and measures of advantage deteriorate directly with remoteness or rurality, and this lower SES is frequently cited as one of the main reasons for poorer health outcomes for people living in rural and remote Australia [12]. However, when looking closely at the way in which Victoria captures emergency presentation data, the dearth of data coverage in regional and rural areas is palpable [13].

For more than 25 years, the Victorian Injury Surveillance Unit (VISU) has been analysing and interpreting injury data across the state. This data informs the development of injury prevention policy, stimulates research, and provides a touchstone for evaluation of preventative initiatives [14]. One of the central datasets that contributes to the VISU is the Victorian Emergency Minimum Dataset (VEMD). The VEMD aggregates emergency presentation data from each of the 38 public hospitals across Victoria that have a 24-h Emergency Department (ED) service [15]. The remaining smaller rural public hospitals with emergency care beds are designated Urgent Care Centres (UCCs). Unlike designated Emergency Departments that can manage most emergencies and have specialised emergency-specific staff available 24 h a day [16], Urgent Care Centres provide first-line emergency care. They have the capacity to perform emergency resuscitation and stabilisation of patients and, where clinically appropriate, prepare patients for transfer to a higher level of care [15]. These facilities also have limited availability of ancillary services, such as X-ray [17]. UCCs currently do not routinely report emergency presentation data to a central database; this data does not appear in the VISU data and was not available when Mitchell, Curtis et al. [9] conducted their study [13]. UCCs, however, provide a key entry point into the Victorian healthcare system for those in smaller rural communities [18]. Therefore, it is imperative to have access to this currently silent data on injured children attending UCCs if we are to genuinely understand any links between socioeconomic disadvantage and childhood injury within the state.

This article will present one element of a larger study using the Rural Acute Hospital Data Register (RAHDaR) to highlight a deficit in our knowledge of childhood injury in the state of Victoria. In doing so, it will inform our developing understanding of the links between socioeconomic disadvantage, rurality and injury. The aim of this current study is to compare the socioeconomic characteristics of children aged 0–14 years with injury-related emergency presentations attending a UCC to those who attend larger Emergency Departments in order to inform gaps in our current understanding of the links between SES and childhood injury in regional Victoria.

With this in mind, the South-West region of Victoria is being used here as an exemplar.

## 2. Materials and Methods

### 2.1. RAHDaR

Victoria has a similar population size to Denmark with more than 6.6 million people living across the 237,269 square kilometre area, with approximately 79% (~5.2 million) residing in an area considered metropolitan and 21% (~1.4 million) in areas considered rural and regional. South West Victoria represents 1.9% (~122,701) of the Victorian population [19]. Participating hospitals in South-West Victoria, Australia, entered into agreements to provide ongoing de-identified episode-level data to the RAHDaR project in return for twice-yearly benchmarking reports [13]. RAHDaR collates data from health services in the region that provide mandated data reports to government that appear in VEMD, as well as data from less-resourced facilities with UCCs that are not currently mandated to report episode-level emergency data. All data from institutions participating in RAHDaR is securely stored within the South West Alliance of Rural Health (an alliance of public health agencies providing information technology services). Data remains the property of the hospitals involved, and only de-identified data leaves the secure servers.

### 2.2. Data Collection

Data were collected from 9 health services: 2 classified by the Department of Health and Human-Services (DHHS) as having an ED service which we have identified as “VEMD” reporting hospitals [15]; and 7 UCCs which we refer to as “Non-VEMD” reporting hospitals [18]. All injury-related emergency presentations—both unintentional and intentional—for children aged 0–14 years recorded at participating facilities between 1 February 2017 and 31 January 2020 and captured in RAHDaR were included in the analysis. Presentations were classified as having a principal injury-related diagnosis using the 10th revision of the International Statistical Classification of Disease and Related Health Problems (ICD-10) Australian modification (ICD-10-AM) are standard across Australia. ICD-10 codes are alphanumeric codes used internationally by health services to classify diagnoses, with every disease, disorder, injury, infection, and symptom having its own code [15]. Data fields were based on the Victorian Emergency Minimum Dataset (VEMD) version 22 (2017–2018) [15]. The de-identified data was extracted from RAHDaR for analysis [17]. Data were aggregated to avoid identification.

In addition to the RAHDaR data above, the principal residence postcode of each child was classified according to the Modified Monash Model (MMM); a health workforce model categorising locations by geographical remoteness and town size [20]. Within the MMM, the South-West region of Victoria is considered rural, with geographical areas classified as either MM3 (Large Rural Town, areas within a 15 km drive of a town between 15,000 to 50,000 residents), MM4 (Medium Rural Town, areas within a 10 km drive of a town with between 5000 and 15,000 residents) or MM5 (Small Rural Town, areas that are or more than 10 km of a town between 5000 and 15,000 residents, and having up to 5000 residents). The four Socio-Economic Indexes for Areas (SEIFA) were also included in the data analysis. The indices included: Index of Relative Socio-economic Disadvantage (IRSD), the Index of Relative Socio-economic Advantage and Disadvantage (IRSAD), the Index of Economic Resources (IER) and Index of Education and Occupation (IEO), and included 2017–2020 population data to adjust for the demographic evolution of the population over the three-year period. SEIFA classifies relative socio-economic advantage and disadvantage as a measure of people’s access to material and social resources, and their ability to participate in society. It is important to appreciate that while an individual’s income is an important element of access and participation, SEIFA data operates at an area level and not at an individual level and therefore tends to represent the characteristics of the area’s residents and the area itself [21].

The four SEIFA indices were used to examine the socio-economic characteristics of the postcode within which the children resided. To achieve this, the various raw scores, deciles, and quintiles for each index were classified according to the home postcode of each child to examine decile or quintile differences. However, the limited size of the dataset at this level required the deciles to be grouped: higher levels of socio-economic disadvantage (“Disadvantage”, Decile 1–5) and higher levels of socio-economic advantage (“Advantage”, Decile 6–10). This allowed a closer examination of the characteristics of relative disadvantage of those presenting to VEMD reporting hospitals compared to those who attended a Non-VEMD reporting hospitals within the sample.

South-West Victoria, Australia is a predominately Caucasian mono-cultural area, with a significant Aboriginal and Torres Strait Islander population (1.8%) when compared to the wider state (0.4%) [19]. Despite this, and in the context of a small sample size, data was not delineated by way of ethnic, cultural, or racial groupings, given the issues of potentially identifying individuals. Likewise, the data presented here has not separated intentional and unintentional injury. Ethical approval for the project required that the researchers were not permitted to report on groupings of individuals of 25 or less given the potential for inadvertent identification.

### 2.3. Data Analysis

Data were cleaned, checked, and analysed using Statistical Package for the Social Sciences (SPSS, Version 25.0). Population figures at the postcode level were used to calculate injury-related emergency presentation incidence rates across the southwest region of Victoria. Presentations at postcode level were divided by the corresponding population estimates for each year for that postcode to obtain standardised injury emergency presentation incidence rates.

Descriptive and inferential statistics were used including statistical and independent sample t-test, ANOVA, and Chi-square (χ^2^) tests to determine if the rate of injuries by postcode differed between the VEMD and Non-VEMD reporting hospitals for children from areas of Disadvantage when compared to children from areas of Advantage. Significance was determined at two-tailed *p* ≤ 0.05.

Ethical approvals were obtained from South West Healthcare Human Research Ethics Committee, Deakin University Human Research Ethics Committee, and Federation University Human Research Ethics Committee (SWH-2019-167567, DUHREC 2019-134, and FUHREC E19-005).

## 3. Results

Across the three-year study period there were a total of 10,137 children aged 0–14 years having a principal injury-related emergency presentation to a participating hospital within South West Victoria. With a population of 22,627 children aged 0–14 years within the study area, this represents an age-standardised rate of 149.8 (56.0 non-VEMD, 93.8 VEMD) per 1000 population (95% CI 148.8 to 149.8). Of the entire sample, 7256 (71.6%) children with injuries were from areas of Disadvantage. Further, it is noted that a higher percentage of younger children who presented with injury were from areas of Disadvantage, while a higher percentage of older children presenting were from areas of Advantage (Table 1). Males had higher numbers of injury presentations than females in both areas of Disadvantage (4115; 56.7%) and areas of Advantage (1680; 58.3%). Within areas of Disadvantage the highest number of presentations were children from Large Rural Towns (MM3)(2947; 40.6%), while children from Smaller Rural Towns (MM5) represented the highest number of presentations within areas of Advantage (2026; 70.3%). Overall, children from areas of Advantage were more likely (2.7% vs. 1.8%) to be transferred to other health services. There was little difference observed between children from areas of Advantage and areas of Disadvantage in terms of the most common injury causes. The mean length of time spent receiving emergency care (Length of Stay) was similar overall between children from areas of Advantage (1.84 h) when compared to their peers from areas of Disadvantage (1.86 h), and when additionally evaluated by facility type.

Interrogating this data further by facility type (Table 2) indicated that a greater number of children from areas of Advantage attended a VEMD reporting hospital than a Non-VEMD reporting hospital (1912 vs. 969), however this may be related to the levels of relative advantage surrounding VEMD hospitals and the size of these populations. Nevertheless, it was observed that a statistically significantly greater proportion of those children attending a Non-VEMD reporting hospital with an injury were from areas of Disadvantage when compared to VEMD reporting hospital (74.3% vs. 70.0%; Table 2).

When comparing the child injury presentation rates from areas of Disadvantage to those from areas of Advantage by postcode across the South West study area, the IRSD, IRSA, IER, and IEO within SEIFA all showed higher mean injury rates (presentations per 1000 children population per year) among children from areas of Disadvantage (*p* = 0.000). (Table 3).

When examining only Non-VEMD reporting hospital injury presentations by postcode, it was noted the highest mean injury rates (presentations per 1000 children population per year) were amongst children with higher levels of relative educational and occupational advantage (IEO, 63.0; Table 4). However, when examining the data within the context of relative socioeconomic disadvantage (IRSD), it was noted that mean injury rates were higher among children from areas of greater Disadvantage (IRSD, 56.2; Table 4).

When considering only VEMD reporting hospitals, higher mean injury rates (presentations per 1000 children population per year) were noted among children from areas of Disadvantage in the context of IRSA, IER and IEO. However, when examining IRSD, mean injury rates were higher among children from areas of Advantage (Table 4). The mean rates of presentation by postcode to VEMD reporting hospitals were approximately double the rates seen at Non-VEMD reporting hospitals.

When examining injury presentations by level of rurality, it was noted that there were higher numbers of children with injury from areas of Disadvantage and residing in an MM3 area, regardless of the type of health service a child attended (VEMD or Non-VEMD reporting hospitals) (Table 5). All children presenting to either a VEMD or Non-VEMD reporting hospital and residing in a postcode classified as MM4 under the Modified Monash Model were considered living in an area of Disadvantage.

When considering IRSD, IRSA, IER, and IEO by level of rurality, there was an inability to calculate differences between children from areas of Advantage and areas of Disadvantage residing in Medium Rural Towns (MM4); all children were classified as being from an area of Disadvantage (Table 6). It was demonstrated that children in areas of Disadvantage within Larger Rural Towns (MM3) experienced the highest mean injury presentation rates by postcode, followed closely by their counterparts from areas of greater Advantage (MM3). Injury-related presentation rates decreased with increasing rurality, with MM4 children having lower presentation rates, and those from Small Rural Towns (MM5) presenting to a health service at the lowest rates. Similar mean presentation rates were observed between both groups from areas of Advantage and areas of Disadvantage within MM5.

## 4. Discussion

This study provides previously missing data on emergency injury presentations of children in rural Victoria, specifically the South West. It demonstrates the importance of including data from smaller hospitals in childhood injury analysis, and the impact this data has on previously held assumptions concerning injury presentation rates, rurality, and level of disadvantage.

One of the significant findings from this study is the rate at which children aged 0–14 years having a principal injury-related emergency presentation presented to a participating hospital within South West Victoria: 149.8 per 1000 population, made up of 93.8 and 56.8 per 1000 population from VEMD and non-VEMD reporting hospitals, respectively. The VISU reports ED (VEMD) presentations for injured children in the same age bracket for the year 2019–2020 at a rate that is slightly lower than we identified in our study (72.05 vs. 93.8 per 1000 population) [14]. Having access to the data from RAHDaR—inclusive of non-VEMD—certainly improves our surveillance of injury and goes some way towards explaining the higher rate of injury amongst children from the South West of Victoria, warranting further consideration [22]. We suggest that studies that examine closely the characteristics of childhood injury, including a delineation of intentional and unintentional injury, as well as examining those factors that influence parental decision making in regard to attending a hospital could help us better understand and address this higher rate.

More than half of injured children were males in both VEMD (56.2%) and Non-VEMD (58.8%) reporting hospitals (Table 1); a considerably lower proportion than that reported by other Australian sources [4,23,24]. There were similar proportions of children injured in the toddler and child years, with an increase in teenage years: not dissimilar to other reports [22,23]. A higher proportion of young children (0–4 years) tended to come from areas of Disadvantage, however, a higher percentage of older children (10–14 years) presenting were from areas of greater Advantage; a finding consistent with another Australian study that found social patterning is different across injury risk behaviours [6]. In addition, when comparing the common injury type we find a surprising degree of consistency across both VEMD and Non-VEMD reporting hospitals.

While no statistically significant difference was identified between the common injury types, there is a noticeable increase in percentage (36.2%) of children presenting to Non-VEMD reporting hospitals with cutting and piercing injuries who reside in areas of Disadvantage. This remains consistent with previous research linking SES and injury, suggesting that the underlying cause of injuries such as this may be the hazards in the living environment that are unique to more regional areas, coupled with a lack of means of protection amongst families of low SES [25,26,27], and warrants closer examination. In addition, an interrogation of the data among the cohort grouped under the VEMD classification ‘Other External Cause’, may provide greater nuanced insights into differences across the cohort of injured children who experience an injury and are classified into this category. However, such an endeavour was beyond the focus of the current study and had the potential to lead to inadvertent identification, as previously discussed.

When comparing overall length of stay, it must be noted that there were some differences between children residing in areas of greater Advantage in the Non-VEMD reporting hospital data. Although not significantly different, comparing departure status of children from areas of Disadvantage and areas of Advantage demonstrated a higher proportion of advantaged children being transferred to another health service. On the other hand, children from areas of greater disadvantage showed a marginally greater propensity to leave the health service prior to full completion of care (“Left at own risk”). Previous studies suggest that those from lower-SES often experience difficult life circumstances that influence the discharge planning, including access to transportation, that may explain higher levels of discharge at own risk [28].

This study identified the incidence of injury-related presentations for children 0–14 years of age was greater overall amongst those from areas of greater Disadvantage (Table 3), with 7256 (71.6%) of children across the entire sample (Table 1) coming from an area classified as disadvantaged. These findings are consistent with the broader international [6,26] and national [4,6,25] research that identifies SES disadvantage as a risk factor for paediatric injury. In a Victorian context, these findings amplify the national study by Mitchell, Curtis at al. [9], who suggested fluctuations in the incidence of childhood injury with regard to SES, but with a slightly higher proportion of injured children residing in areas of socioeconomic disadvantage across all age groups. However, as previously highlighted, in addition to the non-reported data from the Non-VEMD reporting hospitals these authors determined these rates in the absence of some 20% of the available Victorian injury data from the VEMD reporting hospitals. The previously missing Non-VEMD reporting hospital data suggests that our current understanding has significantly under-estimated the degree of injury amongst children from areas of disadvantage within Victoria. The implications of these data gaps for injury prevention strategies that seek to address this disparity have been examined elsewhere [22].

When examining both health service groups it was noted that a higher proportion of children attending a Non-VEMD reporting hospital were classified as being from an area of greater Disadvantage (73.4%) compared to those attending VEMD reporting hospitals (70.0%) (Table 2). The difference in proportions becomes more evident when looking specifically at the Socio-Economic Indexes for Areas (SEIFA) across the entire sample. Children identified as residing in an area of Disadvantage against the IRSD, IRSA, IER and IEO measures showed statistically significantly higher injury rates than their peers from areas of greater Advantage against each measure (Table 3). This finding is consistent with other Australian literature [4,25] that support the links between socioeconomic disadvantage and increased injury rates. 

A close examination of the differences between Advantage and Disadvantage children who attended a VEMD or a Non-VEMD reporting hospital offer, at times, contrasting perspectives with regard to the SEIFA indices. Children residing in areas of Disadvantage who attend a Non-VEMD reporting hospital were, in terms of the IRSD, more likely to experience an injury than their more advantaged peers. While this finding is consistent with the broader Australian literature linking injury and disadvantage [4,6,25], in the context of the IEO measure, children from areas classified as educationally advantaged had higher rates of injury presentation. We postulate that these children’s parents may have higher levels of health literacy which, while it has not prevented injury to their child, makes them more inclined to judge the severity of their child’s injury as requiring medical attention. The role of parental ‘triage’ of the severity of a child’s condition has been described as the major reason for presentation at the emergency department [28]. Further research that examines the decision making processes of parents in regard to attending a health service with their child is needed to better understand the nuances of these considerations.

When examining the same SEIFA indices in the context of children attending only VEMD reporting hospitals the picture is somewhat different. As anticipated, and in line with the literature, injury rates are higher amongst children classified as Disadvantage in the context of IRSA, IER and IEO measures. Interestingly, with regard to IRSD, injury rates were higher amongst children from more advantaged areas. While this finding represents an opportunity for further work, we suggest that this anomaly highlights that injury differences between groups, as observed in the Government reported VEMD hospital data, may not accurately reflect Non-VEMD data and the related assumptions used to inform our understanding of Non-VEMD reporting hospital utilisation. The higher numbers of VEMD reporting hospital presentations within the data may influence the differences that we have highlighted (Table 3 and Table 4), inaccurately suggesting a homogenous difference between levels of Advantage and Disadvantage across service types. This therefore highlights the need for smaller hospital data in shaping our understanding. Central to this argument is the need for accurate injury surveillance, a feature of previous studies [22,29].

It is noteworthy that the rates of presentation (presentations per 1000 children population per year) to VEMD reporting hospitals were approaching double the rates seen at Non-VEMD reporting hospitals. Past research has demonstrated that patterns and rates of injury change with increasing degree of remoteness [12]. Increasing levels of remoteness are consistently associated with a higher prevalence of socioeconomic disadvantage, and people who are socially and economically disadvantaged have poorer health outcomes and increased exposure to health risk [30]. According to the Australian Institute of Health and Welfare (AIHW), the difference between injury rates for children living in remote, rural and the most disadvantaged areas were three times as high as those for children living in the highest SES areas [3]. It would therefore be reasonable to expect that the rates of injury amongst those living in these more rural areas to be highest. However, the VISU data reports ED presentations for injured children in the same age bracket for the year 2019–2020 at a rate that is higher than (72.05 per 1000 population) the rate identified within the non-VEMD data (56.0 per 1000 population) [14]. Therefore, while our study showed that Disadvantage children certainly had higher rates of injury across both MM3 and MM5 areas, a central finding from this study is that higher rates of injury were seen in the less-rural populations (MM3). Further studies that examine the characteristics of injury categorised as “other cause” as well as differentiation between those classified as intentional and unintentional may provide further nuance our understanding of any links between SES and injury amongst this population of children.

Aside from the geographic isolation, rural areas of Victoria are often assumed to experience higher levels of unemployment, and reduced health literacy and education attainment [12]. While the need for more substantial travel to the nearest hospital may contribute to lower presentation rates from those who live in areas classified as Small Rural Towns (MM5), we do not believe it is a significant enough issue on its own to reduce the rate of rural presentation, but is instead one piece in the complex mosaic of childhood injury. Our study has highlighted that those from rural areas with higher levels of education attainment have higher presentation rates at health services, which may, in fact, play a role in the parental assessment of their child’s illness and affect how the family accesses services. Conversely, those who have lower levels of education attainment are thought to have poorer health literacy [31], potentially resulting in poorer recognition of injury severity, which may contribute to the reduced numbers of children appearing in the surveillance data examined from Non-VEMD reporting hospitals.

One possible explanation for the lower presentation rate from children residing in rural areas is the phenomena of ‘rural stoicism’: a perception that rural residents are more stoic than those who live in more metropolitan areas [32]. Reference has been made within the existing literature to suggest rural stoicism as an explanation for the reduced help-seeking behaviours [27]. However, according to previous research there is a lack of evidence to support the position that people living in rural areas are indeed more stoic than their urban counterparts [33]. Instead, it is proposed that differences in stoicism are, in fact, reflections of a lack of access, or lower expectations of healthcare services in rural areas [34]. It is argued that the contemporary notion of ‘rural stoicism’ depicts rural settings as inferior and unprogressive in comparison to urban ones, and that these discourses operate to serve the health agenda of the Government, seeking to disguise the fact that the unique needs of the rural communities are not being addressed [35]. Having access to the nuanced RAHDaR data will offer an avenue to better understand the picture of those in more rural areas who are otherwise missed from our current understanding.

The phenomena of delayed decision to seek treatment, within the context of Acute Coronary Syndrome (ACS), has been shown to be higher amongst those from rural areas as well as lower socioeconomic areas with lower levels of health literacy [35]. Similarly, social cognitive and emotional responses of patients to ACS symptoms played a key role in their decision to seek care, including a mismatch between their perceived symptom expectations and actual experience [36]. While not paediatric nor injury-specific, we believe that further examination of the decision making process that informs parents of injured children to seek treatment begin to will shed light on our understanding of the cohort of people that the Non-VEMD reporting hospitals provide care for.

If metropolitan settings are used to provide the normative healthcare model and outcome measures by which other healthcare settings are measured, we risk perpetuating the ‘deficit discourse’ of rural health that has been highlighted elsewhere as being dominant [37]; the fundamental notion being rural health is in deficit if we make comparisons to urban health as the normative model of healthcare. Rural health outcomes have been noted for many years as being poorer when compared to their urban counterparts. However, as we have shown here, there is diversity in the statistics for urban and rural health that make many comparisons simplistic. Our examination of childhood injury has shown that when we actually have access to the nuanced data we begin to see a different picture relative to normative urban comparisons. This echoes the findings from previous work suggesting that attempts at improving rural health are unlikely to succeed if we persist in ignoring the context which is critical to rural health [37]. Our central thesis is having access to the presentation data of those attending UCCs is essential to our understanding of the links between SES and injury, and needed to begin to better address any deficit.

The comparisons being made here between the proportion of children and families from areas of Disadvantage attending VEMD and Non-VEMD reporting hospitals come with a caution. It is possible that the geographical location of health services alone may have a foreseeable impact upon those differences we have presented here. For example, typically UCCs have limited imaging and other diagnostic capabilities, meaning families and emergency services might tend to bypass the UCCs in favour of VEMD reporting hospital hospitals with the necessary facilities [17], further complicating the presentation and discussion of the data. Irrespective, having access to injury data from UCCs allows us to begin to understand the genuine links that may be present between SES and injury amongst children that have otherwise not been available.

## 5. Conclusions

Our current understanding of childhood injury in regional and rural Victoria is shaped by two factors: firstly, the VEMD reporting hospital data that is available for the VISU database that aggregates injury surveillance information from EDs across the state; and secondly, by way of the urban-rural comparison that has come to normalise the metropolitan model of healthcare as the touchstone which can only ever highlight the deficits of the rural health context. Having access to a significant portion of the injury data that has otherwise been missing from existing databases has confirmed that children from areas of Disadvantage experience higher rates of injury. However, this study has also highlighted that the currently held view that more rural areas are associated with greater levels of disadvantage and will therefore have higher rates of injury is, in fact, not the case. While this study focuses on one geographical area of Victoria, we have no reason to believe that the picture will be any different in other similar areas across the state. The findings presented here may go some way towards re-writing the narrative of rural and urban disparity. This is only possible if we have the nuanced and contextual understanding that the RAHDaR database provides.

## Figures and Tables

**Table 1 ijerph-18-07005-t001:** Comparison of children and areas of Disadvantage and areas of Advantage (according to IRSD) on the basis of: Age, Gender, Presentation by Rurality, SEIFA Indices, Departure Status, Common Injury and Length of Stay.

	VEMD				Non-VEMD				Total			
	Disadvantaged		Advantaged		Disadvantaged		Advantaged		Disadvantaged		Advantaged	
Factor	*n*	(%)	*n*	(%)	*n*	(%)	*n*	(%)	*n*	(%)	*n*	(%)
	4456		1912		2800		969		7256	71.6	2881	28.4
Gender												
- Male (*n* = 5795)	2483	55.7%	1095	57.3%	1632	58.3%	585	60.4%	4115	56.7%	1680	58.3%
- Female (*n* = 4342)	1973	44.3%	817	42.7%	1168	41.7%	384	39.6%	3141	43.3%	1201	41.7%
Age	4456		1912		2800		969		7256		2881	
- 0–4 years (*n* = 3006)	1277	28.7%	475	24.8%	973	34.8%	281	29.0%	2250	31.0%	756	26.2%
- 5–9 years (*n* = 3131)	1370	30.7%	589	30.8%	862	30.8%	310	32.0%	2232	30.8%	899	31.2%
- 10–14 years (*n* = 4000)	1809	40.6%	848	44.4%	965	34.5%	378	39.0%	2774	38.2%	1226	42.6%
Presentations by Rurality	4456		1912		2800		969		7256		2881	
- MM3 (*n* = 3802)	2895	65.0%	817	42.7%	52	1.8%	38	3.9%	2947	40.6%	855	29.7%
- MM4 (*n* = 2714)	1017	22.8%	0	0.00%	1797	64.2%	0	0.00%	2814	38.8%	0	0.00%
- MM5 (*n* = 3521)	544	12.2%	1095	57.3%	951	34.0%	931	96.1%	1495	20.6%	2026	70.3%
SEIFA Index												
- IRSD (*n* = 10,137)	4456	24.1%	1912	27.2%	2800	25.1%	969	24.6%	7256	24.5%	2881	26.3%
- IRSA (*n* = 10,137)	4660	25.3%	1708	24.3%	2862	25.7%	907	23.0%	7522	25.4%	2615	23.8%
- IER (*n* = 10,137)	4370	23.7%	1998	28.5%	2690	24.2%	1079	27.4%	7060	23.9%	3077	28.1%
- IEO (*n* = 10,137)	4962	26.9%	1406	20.0%	2784	25.00%	985	25.0%	7746	26.2%	2391	21.8%
Departure status ^†^	4362		1866		2725		942		7087		2808	
- Home (*n* = 9155)	4029	90.4%	1677	87.7%	2588	92.5%	861	88.9%	6617	91.2%	2538	88.1%
- Admitted (*n* = 526)	301	6.7%	177	9.3%	34	1.2%	14	1.4%	335	4.6%	191	6.6%
- Transfer (*n* = 214)	32	0.8%	12	0.6%	103	3.7%	67	6.9%	135	1.8%	79	2.7%
- Left at own risk (*n* = 194)	94	2.1%	46	2.4%	48	1.7%	6	0.6%	142	2.0%	52	1.8%
- Other (*n* = 25)	0	0.00%	0	0.00%	12	0.4%	13	1.3%	12	0.2%	13	0.5%
- Missing (*n* = 23)	0	0.00%	0	0.00%	15	0.5%	8	0.9%	15	0.2%	8	0.3%
Five most common injuries ^†^	3753		1571		2265		787		6010		2340	
- Fall (<1 m) (*n* = 3632)	1719	47.3%	710	19.5%	896	24.6%	310	8.5%	2612	71.9%	1020	28.1%
- Other external cause (*n* = 2104)	976	46.1%	414	19.6%	557	26.3%	169	8.0%	1530	72.3%	574	27.1%
- Struck/collision object (*n* = 1267)	565	44.4%	228	17.9%	368	28.9%	112	8.8%	932	73.2%	335	26.3%
- Cutting/piercing (*n* = 887)	305	34.3%	125	14.0%	322	36.2%	138	15.5%	627	70.4%	260	29.2%
- Fall (>1 m) (*n* = 460)	188	40.7%	94	20.3%	122	26.4%	58	12.6%	309	66.9%	151	32.7%
Length of stay, Mean hrs (range)	1.87 (0.03–14.57)		1.85 (0.03–14.50)		1.83 (0.03–18.32)		1.86 (0.02–16.07)		1.86 (0.03–18.32)		1.84 (0.02–16.07)	

VEMD: Victorian Emergency Minimum Dataset; non-VEMD health services not included in VEMD; MM3: Large Rural Town; MM4: Medium Rural Town; MM5: Small Rural Town; IRSD: Index of Relative Socio-economic Disadvantage; IRSAD: Index of Relative Socio-economic Advantage and Disadvantage; IER: Index of Economic Resources; IEO: Index of Education and Occupation. ^†^—As per VEMD data definitions [15].

**Table 2 ijerph-18-07005-t002:** Comparing Non-VEMD and VEMD Reporting Hospital presentations for children from areas of Disadvantage and Advantage (according to IRSD).

	Disadvantage		Advantage			
	Number	Percentage	Number	Percentage	χ^2^-Test	*p*-Value
Non-VEMD	2800	74.3%	969	25.7%	21.674	0.000
VEMD	4456	70.0%	1912	30.0%		
Total	7256	71.6%	2881	28.4%		

**Table 3 ijerph-18-07005-t003:** Injury rate differences between Advantage and Disadvantage throughout the study area (all health services).

	Advantaged	Disadvantaged	df	*t*-Test	*p*-Value
	Rate ^†^	Rate ^†^			
IRSD	155.0	161.6	10,134	9.965	0.000 *
IRSA	152.1	162.4	10,134	15.158	0.000 *
IER	152.5	162.8	10,134	15.994	0.000 *
IEO	146.7	163.7	10,134	24.812	0.000 *

^†^ per 1000 children population per year. * *p* < 0.01.

**Table 4 ijerph-18-07005-t004:** Injury differences between Advantage and Disadvantage that attend VEMD and Non-VEMD reporting hospitals.

	VEMD					Non-VEMD				
	Advantaged	Disadvantaged				Advantaged	Disadvantaged			
	Rate ^†^	Rate ^†^	df	*t*-Test	*p*-Value	Rate ^†^	Rate ^†^	df	*t*-Test	*p*-Value
IRSD	111.5	107.7	6376	−2.276	0.023 *	53.3	56.2	3815	2.209	0.027 **
IRSA	104.0	110.5	6376	3.834	0.000 *	56.4	55.1	3815	−0.936	0.349
IER	105.2	110.4	6376	3.187	0.001 *	55.2	55.6	3815	0.297	0.766
IEO	95.0	112.9	6376	10.219	0.000 *	63.0	53.2	3815	−7.016	0.000 *

^†^ per 1000 children population per year. * *p*<0.01, ** *p*<0.05.

**Table 5 ijerph-18-07005-t005:** Comparing Non-VEMD and VEMD reporting hospital presentations for children from areas of Disadvantage and Advantage in the context of Rurality.

Rurality	VEMD		Non-VEMD		Total	
	Disadvantaged	Advantaged	Disadvantaged	Advantaged	Disadvantaged	Advantaged
	*n*(%)	*n*(%)	*n*(%)	*n*(%)	*n*(%)	*n*(%)
MM3	2895 (78.0)	817 (22.0)	52 (57.8)	38 (42.2)	2947 (77.5)	855 (22.5)
MM4	1017 (100.0)	0 (0)	1797 (100.0)	0 (0)	2814 (100.0)	0 (0)
MM5	544 (33.2)	1095 (66.8)	951 (50.5)	931 (49.5)	1495 (34.7)	2026 (65.3)

**Table 6 ijerph-18-07005-t006:** Total injury rate differences between levels of rurality according to Advantage and Disadvantage.

	MM3	MM4	MM5			
Advantage	Rate ^†^	Rate ^†^	Rate ^†^	df	*t*-Test	*p*-Value
– IRSD	171.7	-	147.9	2878	16.378	0.000 *
– IRSA	158.2	-	150.4	2612	4.499	0.000 *
– IER	173.5	-	145.2	3073	18.297	0.000 *
– IEO	173.5	-	145.9	2388	2.859	0.004 *
**Disadvantage**	**Rate ^†^**	**Rate ^†^**	**Rate ^†^**	**df**	**ANOVA**	***p*-value**
– IRSD	174.0	157.6	144.5	2, 7519	1224.907	0.000 *
– IRSA	176.2	157.6	141.0	2, 7253	809.219	0.000 *
– IER	174.1	157.6	148.7	2, 7057	645.853	0.000 *
– IEO	175.2	157.6	147.4	2, 7743	933.041	0.000 *

^†^ per 1000 children population per year: MM3: Large Rural Town; MM4: Medium Rural Town; MM5: Small Rural Town. * *p* < 0.01.

## Data Availability

Due to confidential nature of the data, research data cannot be shared.

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
