# Peer review of "The Socioeconomic Characteristics of Childhood Injuries in Regional Victoria, Australia: What the Missing Data Tells Us"

_ijerph, 2021, doi:10.3390/ijerph18137005_

Round 1
Reviewer 1 Report
I commend the authors for an important study that points to injury disparities for disadvantaged children in Victoria, Australia. Low social economic status has been linked in multiple studies to increased rates of injury as well as increased morbidity and mortality from injury. An understanding of this in the Australian context is helpful. There are some additional clarifications needed from the authors:
- A more clear definition of disadvantaged vs. advantaged is needed. What income levels constitute poverty?
- There were about 10,000 children with injuries in this study. What is the denominator? How do rates of injury in Victoria compare to the rest of Australia?
- Did the authors collect any racial/ethnicity data and were there any racial/ethnic disparities apparent in the data?
- With regards to # 2, there is data that Indigenous Australian Aboriginal children suffer a disproportionately high burden of injury. Was this reflected in the data the authors collected? If so additional clarification and contextualization of this information will be important
- Were there any particular injury type differences noted in rural vs. urban populations especially injuries related to farming, outdoor activities, hunting etc.?
- The authors describe differences based on city size but no numbers to explain what constitutes an MM3 vs MM4 vs. MM5 size city.
Author Response
Dear Reviewers
We wish to thank you for your considered feedback on this paper. Your feedback has certainly strengthened the paper and your expert advice has and will continue to drive our research program in this area forward.
Many thanks
Blake

Reviewer 2 Report
Thank you for the opportunity to review manuscript ID ijerph-1243415 entitled ‘The socioeconomic characteristics of childhood injuries in regional Victoria, Australia: What the missing data tells us’ which was submitted for potential publication to the International Journal of Environmental Research and Public Health.
This study reports the socio-economic characteristics of childhood injuries in regional Victoria, through comparison of emergency department presentations to presentations at urgent care centres, which the authors note, are often missing from routinely collected data. This study is well-written and fills a gap in understanding of child injury (and its prevention) and the impact of determinants of health, such as socioeconomic status and rurality. I have some general suggestions and then some specific comments which I hope the authors will find useful. I wish the authors continued luck in getting their study published.
General comments
Might be worth adding a sentence or two to the introduction to explain Victoria to international readers? I.e., location within Australia, population etc. This is an international journal after all.
See also relevant literature which is not currently referenced which explores child injury-related mortality in Australia by rurality and socio-economic status: https://www.mdpi.com/2227-9067/8/1/5
Does this study explore unintentional and intentional injury? Worth explaining/clarifying in abstract and then in methods. Line 258 – first mention of unintentional
Throughout the results the grammar is odd given the labels of the groups – perhaps areas of Disadvantage, areas of Advantage is better grammatically and could be run throughout? This continues into the discussion and flips between areas of disadvantage, disadvantage and disadvantaged. Check grammar throughout.
The ‘other external cause’ category is extremely large, can you speak to why this was grouped and add more to the methods. This is also a limitation that should be noted, hard to derive much from these insights, without knowing more about the mechanisms within this broad grouping. To this end, I was surprised to not see more analysis by injury mechanism, given this is really valuable information for prevention.
Tables should be able to stand alone. Add table notes ie below Table 1 to explain acronyms used such as IRSD, MM3 etc. Check all tables considering this point.
Specific comments
Abstract
Time period of the study needs to be noted somewhere – probably methods
Line 16 – would add , Australia after Victoria so readers are again made aware of the context of the study (in addition to the title)
Introduction
Line 33 – would rephrase first sentence to be about the period of childhood and not childhood injury – doesn’t read quite right at the moment
Lines94-97 – will you be testing this assumption in this or the broader work?
Materials and Methods
Line 113 – aged 0-14 years of age is a bit redundant. Aged 0-14 or 0-14 years of age would be fine, don’t need both
Results
Line 169-170 – the words to be need to be added prior to transferred
Line 176 – Table 1 header – SEIF should be SEIFA
Table 2 – why did you not run the same X2 analysis of differences across Table 1? I would think it would add more value
The order the information is presented in seems odd, would you not start with Table 3 – ie all health services and then explore within that?
What do the – to the left of the Variables under Advantage and Disadvantage in Table 6 represent?
Discussion
Line 235 – 56.2% and 58.8% are not a majority, they are just over half
Line 244 – expand on the similarities re injury mechanism (type) you identified. Doesn’t need to be much, a sentence or two
Line 357 – check clarity of expression / being to or will?
Data availability statement- add or remove standard template text
Author Response

(The authors gave the same response as above.)

Round 2
Reviewer 2 Report
Thanks to the authors it is an improved manuscript. I still feel there is limited value to this paper without more interrogation of the injury mechanism to guide prevention efforts. It seems quite cynical to say this will be dealt with in future papers. Additionally, the ethical restrictions seem quite onerous, I am used to concealing small counts less than 5 but 25 seems to make meaningful analysis within a state extremely difficult.
See some addition issues:
Abstract - data were not data was - data are plural please check throughout
Check formatting of references in text. Some inside punctuation and some outside. Some with spaces, some placed immediately after the word.
In addition to noting some of the challenges with small case numbers and missing variables (ie ethnicity, First Nations children) in the methods, it would be worth adding a strengths and limitations section to the discussion to further note this with recommendations for future work in this space.